# Effect of Time-of-Day-Exercise in Group Settings on Level of Mood and Depression of Former Elite Male Athletes

**DOI:** 10.3390/ijerph16193541

**Published:** 2019-09-22

**Authors:** Khadijah Irandoust, Morteza Taheri, Hamdi Chtourou, Pantelis Theo Nikolaidis, Thomas Rosemann, Beat Knechtle

**Affiliations:** 1Department of Sport Sciences, Imam Khomeini International University, Qazvin 34148-96818, Iran; irandoust@soc.ikiu.ac.ir (K.I.); m.taheri@soc.ikiu.ac.ir (M.T.); 2Activité Physique, Sport et Santé, UR18JS01, Observatoire National du Sport, Tunis 1003, Tunisia; h_chtourou@yahoo.fr; 3Institut Supérieur du Sport et de l’éducation physique de Sfax, Université de Sfax, Sfax 3000, Tunisia; 4Exercise Physiology Laboratory, 18450 Nikaia, Greece; pademil@hotmail.com; 5Institute of Primary Care, University of Zurich, 8091 Zurich, Switzerland; thomas.rosemann@usz.ch; 6Medbase St. Gallen Am Vadianplatz, 9001 St. Gallen, Switzerland

**Keywords:** time-of-day, mood, depression, elite athlete

## Abstract

Since the prevalence of depression is high among athletes at the end of their athletic career, the purpose of this study was to investigate the effect of time-of-day-exercise in group settings on the level of the mood and depression of former elite male athletes.Out of 187 volunteers referring to the sports counseling clinic, 71 retired male athletes who had a national championship record were randomly divided into two morning and evening exercise groups. The inclusion criteria were severe depression (high score in the Beck Depression Inventory-II), the age range of 50 to 60 years, the absence of metabolic syndrome, and the body mass index (BMI) between 28 and 35. All body composition variables were measured using body composition analysis (In Body 320; Korea). The second stage was the collection of data after three months (completion of the training protocol). After data collection, independent and dependent t-tests were used to analyze the data. The results indicated that both groups had a significant improvement in depression compared to the pre-test (*p* ≤ 0.05), while there was no significant difference between the two groups (*p* ≥ 0.05). The overall conclusion is that exercise at different times of the morning or evening can improve the psychological state and reduce depression.

## 1. Introduction

According to the World Health Organization, after heart disease, depression is the second most commonly occurring disease in the current century [1]. This wide-spread disease, regardless of background, nationality, gender, or health status, may occur for all age groups. To this end, one of the concerns of researchers is to find a solution to prevent or improve depression and other psychological problems, such as mood in the vulnerable populations. Nowadays, championship sport has a lot of attractions for its audiences so that its athletes become legends for fans [2]. In this regard, the long-term health impact of elite competitive sports participation is receiving increasing public attention.

Interestingly, after the end of athletes’ careers and the waning of fans’ attention, many psychological problems, such as frustration and depression are caused.

Transitioning from professional sport to a sedentary profession may lead to low social interactions [3].

Based on research evidence, regular sports participation is associated with a wide array of social and psychological benefits [4,5]. However, the physical and psychological benefit of sports participation dissipates following sport cessation, and elite athletes who become inactive after retirement from sport face the risk of developing psychological problems. So, it seems logical to retain a physically active lifestyle after retirement.

Although a myriad of environmental factors, such as nutritional disturbances [6] and sleep disorders [7], may play a key role in psychological disorders, it seems that providing the former athletes with the involvement in a sports atmosphere (favorite sports group setting) can be effective in improving the condition.

It is well documented that former elite athletes tend to retain an active lifestyle in their senior years. So, it is important to find out the best type of exercise that fits this goal. It was reported in a study that 60% of former elite athletes retain a physically active lifestyle throughout their adulthood, whereas only 17% of controls do the same [8]. Therefore, finding a strategy to reduce the consequences of psychological problems is of utmost importance. Although the beneficiary role of sport in reducing mental disorders and in reinforcing positive emotions is well known [9], it isimportant to know what types of exercises can be more useful in reducing the consequences of such problems.

Overweight and obesity, non-contagious diseases, and mental disorders are post-championship complications [10]. By studying past research, we found that there is a relationship between physical activity, mood, and psychological features [11]. This notion has been accepted in every way that regular exercise can improve these traits. According to a previous study, weight gain in middle age can have metabolic effects on the body, with consequences such as depression, sleep disturbance, and other psychological problems [12]. Therefore, it is worth considering strategies for improving the health of former athletic champions to avoid the occurrence of psychological complications. Doing exercise in group settings can boost motivation and can be a good way to achieve this goal [13]. Given the fact that there is always a question about the practice time for better effectiveness, it is important to consider this issue. One of the uncertainties that has always been of interest among health researchers deals with the effectiveness of time-of-day-exercise on physical and mental health [14]. In some studies, it has been shown that exercise in the morning improves the psychological state [15,16]. According to these results, exercise in the morning not only activates the metabolic and physiological activities but also boosts energy that leads to improved mental activity. Furthermore, a morning sweat may also lead to better mental health and productivity throughout the day, since exercise is great for reducing stress.

However, there is also research that has proven the effectiveness of evening exercises on well-being [17,18]. This has led researchers to explore psychological treatments focused on resetting the body’s clock using such techniques as time-of-day-exercises. Although athletes participate in high levels of physical activities during their sports careers, they may not transfer this into regular exercise after retirement.

Generally, given the facts mentioned above, this study was planned to examine the effect of training in the morning and afternoon hours on the mood and depression of former elite male athletes. We hypothesized that both morning and evening training sessions have a beneficial effect on the mood and depression of former elite male athletes; however, this positive effect could be better after the morning rather than evening training.

## 2. Materials and Methods

### 2.1. Ethical Approval

The study protocol was approved by the ethical committee of the Imam Khomeini International University under the No. 17628.

### 2.2. Subjects

Out of 187 volunteers referring to the sports counseling clinic, 71 former elite male athletes who had a national championship record (at least referred to 15 years ago) were randomly divided into twogroups—morning (8:00 a.m.) and evening (6:00 p.m.) exercise groups. Three subjects were removed due to absence atthe training sessions and onewas absent in the post-test phase. The inclusion criteria were severe depression (score of 14–19 in the Beck Depression Inventory-II (BDI-II)), the age range of 50 to 60 years, the absence of metabolic syndrome, the BMI between 28 and 35, no use of medication, no presence of disease, and a lack of regular physical activity in the last six months. BDI-IIitems are rated on a four-pointscale ranging from 0 to3 based on the severity of each item. The maximum total score is 63 (0–13, minimal depression; 14–19, mild depression; 20–28, moderate depression; 29–63, severe depression). The Profile of Mood States (POMS) was used to evaluate the mood. It is a 30 items questionnaire that is scored on a 5-point Likert-type scale anchored with “Not at all” and “Extremely”. The POMS provides six subscales: (1) anger (ANG) e.g., grouchy, furious; (2) confusion (CON) e.g., muddled, forgetful; (3) depression (DEP) e.g., sad, unworthy; (4) fatigue (FAT) e.g., tired, sluggish; (5) tension (TEN) e.g., nervous, anxious; and (6) vigor (VIG) e.g., lively, active. The POMS has demonstrated adequate reliability with Cronbach’s alpha ranging from 0.93 to 0.95 in previous studies [19,20]. Total Mood Disturbance (TMD) is calculated by summing the totals for the negative subscales and then subtracting the totals for the positive subscales:

TMD = (TEN + DEP + ANG + FAT + CON) – (VIG)
(1)


Subjects participated in the exercise protocol for 12 weeks, 3 times a week lasting 60 min in each training session with 80% maximum heart rate. The exercise protocol included: three phases of warm-up (10 min of stretching exercises), the main program (jogging for 30 min), and cooling-down (5 min of stretching movements). The exercise intensity was evaluated and controlled by Polar Electro, Kempele, Finland.

All measures were conducted in the Dr Irandoust Sporting Counseling Center in 2018 (Jan–March). The exercise protocol and nutritional habits of subjects were controlled by telephone, counseling in the clinic, or self-reporting. The nutrition program was prepared by the nutrition expert based on the number of calories they received each day, which was based on their calculated basal metabolic rate (BMR). The second stage was the collection of data after 3 months (completion of the training protocol). All body composition variables were measured using body composition analysis (In Body 320; Korea) [21].

### 2.3. Statistical Analysis

Results of all parameters are presented as a mean ± standard deviation (SD). Data analyses were carried out using the SPSS v21.0 software (SPSS Inc., Chicago, IL, USA) and Microsoft Excel 2010 (Microsoft Corp., Redmont, WA, USA). For all parameters, the normality of the distributions was tested using the Shapiro–Wilk test. Pre- to post-training and between-groups comparisons were performed using independent and dependent t-tests to analyze the data.The significant difference was set at an alpha level of *p* ≤ 0.05.

## 3. Results

The sociodemographic characteristics of participants are shown in Table 1. Comparing the data of TMD and its components and BDI in both groups, nosignificant difference at baseline and after the intervention period was observed (*p* > 0.05). As indicated in Figure 1 and Figure 2, BDI and TMD improved significantly in both groups after compared to before the intervention (*p* ≤ 0.05). It was also shown that body fat percentage (BFP) of both groups decreased after compared to before the intervention for both groups (*p* ≤ 0.05; 6% and 7% for morning exercise (ME) and evening exercise (EE) groups). The results of the paired *t*-test indicated that both groups had a better improvement in depression compared to pre-test (*p* ≤ 0.05), while there was no significant difference between the two groups (*p* > 0.05). As shown in Table 2, anger, confusion, fatigue, tension, and vigor were also better in both groups (*p* ≤ 0.05) after compared to before the intervention without between-groups differences (*p* > 0.05).

## 4. Discussion

Regarding the importance of mental well-being and mental health in middle age, especially those athletes who have been very popular in the past and have been marginalized due to the end of the championship period, this study examined the effects of two time-of-day training sessions, i.e., in the morning and in theevening, on the mood and depression levels of former male athletes. As observed, TMD was significantly improved following both training programs in the morning and the evening. In addition, all components of mood such as anger, confusion, depression, fatigue, tension, and vigor had a significant improvement following morning and evening aerobic exercise programs; however, there was no significant difference between effectiveness of the morning and the evening exercise programs. According to previous studies, there is a direct correlation between body fat percentage (especially visceral fat) and depression [22,23]. As Voglegean et al. [24] stated, symptoms of depression can be associated with an increase in abdominal fat accumulation. Since both groups of this study experienced a significant decrease in BFP following exercise intervention, it sounds reasonable that their levels of depression decreased. Based on existing theories, physical activity can reduce stress and depression. The explanation for this phenomenon is the increase in serotonin and norepinephrine levels during exercise activities, which reduces depression [25]. On the other hand, adaptations resulting from enhanced androgen releasing and reduced cortisol levels in people with physical activity can lower depression [26]. Generally, exercise is considered as an appropriate non-invasive and low-cost mechanism [27,28,29,30]. Therefore, it is a good way to promote health and reduce depression and mood disturbance in former elite male athletes.

Since the results show that exercise responses are not significantly different in both measures following exercise in the morning and evening, the researchers cannot necessarily conclude that exercise in the morning or evening is better than the other one. At this point, we can only conclude that the effects of the two appear to be the same, and we certainly have to do more work to determine the potential mechanisms for the beneficial effects of exercise training performed at these two time-points. Since various sports exercises can have different effects [31], it is recommended to use diverse exercise protocols with different intensity in upcoming research. Indeed, as indicated in a previous literature review [32,33] and in previous studies that utilized strength training [34,35] and a taper period after training at a specific time-of-day [36], significant differences in physical performances between training in the morning or in the evening hours have been observed. Since morning and evening times for exercise showed no significant difference in improving the psychological state of the subjects, it is suggested that a greater number of participants be applied in future studies and also more control over the intervening variables be conducted.

One limitation of the study refers to the low number of participants and also not performing the chronotype evaluation (for accurate examination of the effect of two kinds of morning and evening exercises). So, it is suggested that they will be obviously considered in future studies. Considering a control group is another limitation to the present study and is highly recommended in future studies. Another limitation deals with considering just the two circadian stages (namely, morning versus evening exercise), so it should be more focused on different aspects of circadian stages. Based on the obtained results, it is recommended to motivate retired elite athletes to contribution in sport activities. Additionally, considering different circadian stages between the time of awakening and bedtime would allow the characterization of the circadian response rhythm and thus more research is needed to make crucial decisions.

## 5. Conclusions

Whilst the number of studies investigating the psychological benefits of sport participation for retired elite athletes was not large, there was a general consensus that there are many psychological and social health benefits associated with participation in sport for retired elite athletes. The overall conclusion is that exercise at different times of the morning or evening can improve the mood status and reduce depression. Therefore, retired elite athletes should maintain regular exercise (morning or evening) in order to maintain their mental health and reduce the risk of psychological problems in the later years of life.

## Figures and Tables

**Figure 1 ijerph-16-03541-f001:**
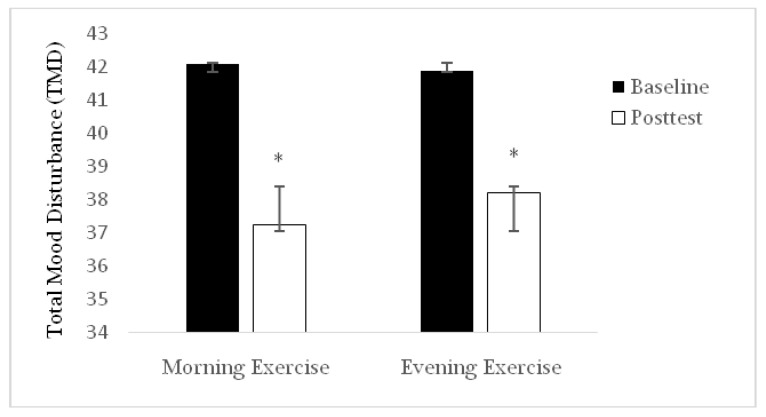
Total mood disturbance (TMD) recorded before (baseline) and after (post-test) the training period in the morning exercise (ME) and the evening exercise (EE) training groups. * Significant difference compared to baseline.

**Figure 2 ijerph-16-03541-f002:**
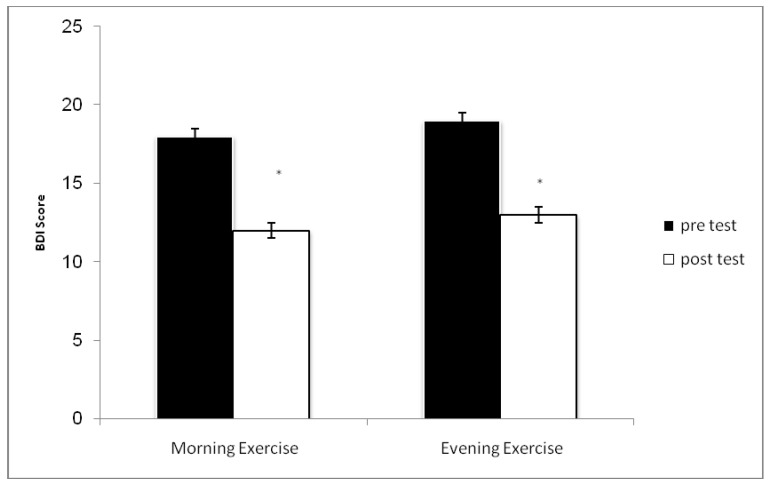
The Beck Depression Inventory-II(BDI-II) score recorded before (baseline) and after (post-test) the training period in the morning exercise (ME) and the eveningexercise (EE) training groups. * Significant difference compared to baseline.

**Table 1 ijerph-16-03541-t001:** Sociodemographic characteristics of participants.

**Age (year)**	46.2 ± 2.1	
**Weight**	85.3 ± 3.7	
**WHR**	0.94 ± 0.01	
**Variables**	**Classification**	**Percentage**
**Income**	Low (≤2,000,000 T)	9%
	Medium (2,000,0000–4,000,000 T)	67%
	High (≥4,000,000 T)	24%
**Level of education**	Diploma	11%
	Associate	4%
	Bachelor	35%
	Master	26%
**Occupation**	Employed	78%
	Self-employed	22%
**Marital status**	Single	4%
	Married	81%
	Widowed	8%
	Divorced	7%

T: Tomans: Iran Currency.

**Table 2 ijerph-16-03541-t002:** The Profile of Mood States parameters recorded before (Baseline) and after (Post-test) the training period in the morning exercise (ME) and the eveningexercise (EE) training groups.

Mood States Parameters	ME (*n* = 34)		*t*	*p*	EE (*n* = 33)		*t*	*p*
	Baseline	Post-Test			Baseline	Post-Test		
**Anger**	8.21 ± 1.5	6.25 ± 1.6	7.14	0.001	8.51 ± 1.5	6.11 ± 1.9	7.25	0.001
**Confusion**	9.72 ± 1.7	8.85 ± 1.8	2.81	0.008	10.21 ± 1.5	8.35 ± 1.6	6.8	0.001
**Depression**	13.52 ± 2.1	11.3 ± 2.4	5.58	0.001	12.9 ± 1.8	10.9 ± 1.5	7.65	0.001
**Fatigue**	16.27 ± 3.2	14 ± 2.6	4.48	0.001	15.1 ± 2.2	13.6 ± 1.9	4.53	0.001
**Tension**	10.31 ± 1.9	8.1 ± 1.2	10.69	0.001	9.3 ± 2.1	8 ± 1.5 ± 2.2	3.13	0.003
**Vigor**	6.45 ± 1.4	4.8 ± 1.0	9.33	0.001	6.2 ± 1.3	5.1 ± 1.1	5.75	0.001

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
