# Peer review of "Effect of Time-of-Day-Exercise in Group Settings on Level of Mood and Depression of Former Elite Male Athletes"

_ijerph, 2019, doi:10.3390/ijerph16193541_

Round 1

Reviewer 1 Report

Interesting study. Could benefit by more thorough review of literature. As well the methods need more clarity. For example, apart from exercising at 80% HR max, there is no description of the actual exercise program

Author Response

Reviewer 1:

Interesting study. Could benefit by more thorough review of literature. As well the methods need more clarity. For example, apart from exercising at 80% HR max, there is no description of the actual exercise program

Answer: Corrections made as suggested. Please see changes made in the text.

Reviewer 2 Report

This is an original study which aimed to compare the effect of time-of-day-exercise in former elite male athletes. It is an interesting issue; however some points need to be enlightened.

Introduction

I believe that there is a lot of other factors which can influence in the incidence of depression in former athletes besides of stop training, such as environmental stimuli, social interaction, nutritional aspects, a goal in life. These factors must be included and exploring in the introduction.

It is not a rational to compare morning and afternoon interventions. What the difference between them? What is the hypothesis? Why these two groups could show different responses?

Methods

It is necessary to explain exactly the hour of interventions. If consider circadian cycle of hormonal responses, for example, 7:00 am is different of 11:30 am.

The subjects have used some type of medication?

What is the mean of age of the subjects?

How long time ago did they compete?

It is necessary to describe the methodological aspects of intervention of exercise. Which type of aerobic exercise did they practice?

If the aim is to compare the changes in depression and POMS scale between groups, the statistical analyses correct is a Repeated measure ANOVA or a delta (pos-pre) comparison with a t-test.

It was evaluated gaussianity and homocesdasticity?

Results

It is missed a table with sociodemographic sample characteristic.                          

It is important to inform If data of graphics were expose in means and standard deviation.

Discussion

Authors should discuss the effect of time of exercise in depression; To compare the results with previous studies; The limitations of the study is scarce, for example is important to explore the lack of control group.

Author Response

Reviewer 2

This is an original study which aimed to compare the effect of time-of-day-exercise in former elite male athletes. It is an interesting issue; however, some points need to be enlightened.

Introduction

I believe that there is a lot of other factors which can influence in the incidence of depression in former athletes besides of stop training, such as environmental stimuli, social interaction, nutritional aspects, a goal in life. These factors must be included and exploring in the introduction.

Answer: Corrections made as suggested. Please see changes made in the text.

It is not a rational to compare morning and afternoon interventions. What the difference between them? What is the hypothesis? Why these two groups could show different responses?

Answer: Corrections made as suggested. Please see changes made in the introduction section.

Methods

It is necessary to explain exactly the hour of interventions. If consider circadian cycle of hormonal responses, for example, 7:00 am is different of 11:30 am.

Answer: Corrections made as suggested. Please see changes made in the text.

The subjects have used some type of medication?

Answer: All participants didn’t use any type of medication. This information has been added in the revised version. Please see changes made in the text.

What is the mean of age of the subjects?

Answer: This information has been added in the revised version. Please see changes made in the text.

How long time ago did they compete?

Answer: They compete for at 15 years. This information has been added in the revised version. Please see changes made in the text.

It is necessary to describe the methodological aspects of intervention of exercise. Which type of aerobic exercise did they practice?

Answer: The following paragraph has been added:Subjects participated in exercise protocol for 12 weeks, 3 times a week lasting 60 min in each training session with 80% Maximum heart rate. The exercise protocol included three phases of warm up (10 min: stretching exercises) - the main program (jogging for 30 minutes) of cooling down (stretching movements, 5 min). The exercise intensity was evaluated and controlled by Polar Electro, Kempele, Finland.” Please see changes made in the revised version.

If the aim is to compare the changes in depression and POMS scale between groups, the statistical analyses correct is a Repeated measure ANOVA or a delta (pos-pre) comparison with a t-test.

Answer: The reviewer is right. A two-way ANOVA is appropriate for the comparison of the present study. However, for the aim of the present study, we need only comparisons pre- to post- training or between groups (the interaction group × pre to post training is not in the aim of the present study). This way we selected a t-test.

It was evaluated gaussianity and homocesdasticity?

 Answer: For all parameters, normality of the distributions was tested using the Shapiro-Wilk test. This information has been added in the revised version. Please see changes made in the text.

Results

It is missed a table with sociodemographic sample characteristic.                          

Answer: This table has been added in the revised version. Please see changes made in the text.

It is important to inform If data of graphics were expose in means and standard deviation.

Answer: This information has been added in the revised version. Please see changes made in the text.

Discussion

Authors should discuss the effect of time of exercise in depression; To compare the results with previous studies; The limitations of the study is scarce, for example is important to explore the lack of control group.

Answer: Correction made as suggested. Please see changes made in the text.

Reviewer 3 Report

1. The introduction is too weak and too short to provide enough information to understand the background and the rationale of this study. 

2. The materials and methods should be revised. The subjects should provide information about the subjects (e.g., demographic information, why only male subjects). BDI, PMOS, and other information can be in Measures. Procedures and Statistical Analysis should be separate. 

3. There must be detailed information about the results and figures. 

4. There should be limitations and suggestions for future research.

5. Conclusions can't be only one sentence. 

Author Response

Reviewer 3

The introduction is too weak and too short to provide enough information to understand the background and the rationale of this study. 

Answer: Additional information has been provided in the revised introduction. Please see changes made in the revised version.

The materials and methods should be revised. The subjects should provide information about the subjects (e.g., demographic information, why only male subjects). BDI, PMOS, and other information can be in Measures. Procedures and Statistical Analysis should be separate. 

Answer: This information has been added in the revised version. Please see changes made in the text.

There must be detailed information about the results and figures. 

Answer: Correction made as suggested. Please see changes made in the text.

There should be limitations and suggestions for future research.

Answer: Correction made as suggested. Please see changes made in the text.

Conclusions can't be only one sentence.

Answer: Correction made as suggested. Please see changes made in the text.

Round 2

Reviewer 2 Report

The authors have included important points and informations in  the manuscript . So, I will accept . Congratulations!

Reviewer 3 Report

Thank the authors for updating the manuscript with reviewers' comments.